# Specialized Pro-Resolving Mediators and the Lymphatic System

**DOI:** 10.3390/ijms22052750

**Published:** 2021-03-09

**Authors:** Jamie D. Kraft, Robert Blomgran, Iben Lundgaard, Marianne Quiding-Järbrink, Jonathan S. Bromberg, Emma Börgeson

**Affiliations:** 1Wallenberg Laboratory, Department of Molecular and Clinical Medicine, Institute of Medicine, Sahlgrenska Academy, University of Gothenburg, 40530 Gothenburg, Sweden; jamie.kraft@wlab.gu.se; 2Wallenberg Centre for Molecular and Translational Medicine, University of Gothenburg, 40530 Gothenburg, Sweden; 3Department of Biomedical and Clinical Sciences, Division of Inflammation and Infection, Faculty of Medicine and Health Sciences, Linköping University, 58185 Linköping, Sweden; robert.blomgran@liu.se; 4Department of Experimental Medical Science, Lund University, 22100 Lund, Sweden; iben.lundgaard@med.lu.se; 5Wallenberg Centre for Molecular Medicine, Lund University, 22100 Lund, Sweden; 6Department of Microbiology and Immunology, Institute of Biomedicine, Sahlgrenska Academy, University of Gothenburg, 40530 Gothenburg, Sweden; marianne.quiding-jarbrink@microbio.gu.se; 7Department of Surgery, University of Maryland School of Medicine, Baltimore, MD 21201, USA; JBromberg@som.umaryland.edu; 8Department of Microbiology and Immunology, University of Maryland School of Medicine, Baltimore, MD 21201, USA; 9Center for Vascular and Inflammatory Diseases, University of Maryland School of Medicine, Baltimore, MD 21201, USA; 10Marlene and Stewart Greenebaum Comprehensive Cancer Center, University of Maryland, Baltimore, MD 21201, USA; 11Region Vaestra Goetaland, Department of Clinical Physiology, Sahlgrenska University Hospital, 41345 Gothenburg, Sweden

**Keywords:** specialized pro-resolving mediators (SPMs), lymphatics, resolution of inflammation, lymphoid organs, atherosclerosis, inflammatory bowel disease, dry eye disease

## Abstract

Diminished lymphatic function and abnormal morphology are common in chronic inflammatory diseases. Recent studies are investigating whether it is possible to target chronic inflammation by promoting resolution of inflammation, in order to enhance lymphatic function and attenuate disease. Resolution of inflammation is an active process regulated by bioactive lipids known as specialized pro-resolving mediators (SPMs). SPMs can modulate leukocyte migration and function, alter cytokine/chemokine release, modify autophagy, among other immune-related activities. Here, we summarize the role of the lymphatics in resolution of inflammation and lymphatic impairment in chronic inflammatory diseases. Furthermore, we discuss the current literature describing the connection between SPMs and the lymphatics, and the possibility of targeting the lymphatics with innovative SPM therapy to promote resolution of inflammation and mitigate disease.

## 1. Introduction

The lymphatic system is an immense network responsible for maintaining tissue homeostasis through the transport of interstitial fluid, dietary lipids, macromolecules and immune cells [1]. The interstitial fluid absorbed by the lymphatics, known as lymph, was initially described by Hippocrates (460–377 B.C.) as “white blood” [2,3]. In contrast to the blood vasculature, the lymphatics have historically been less well-studied until the 1990s, when lymphatic endothelial cell markers were identified that finally permitted visualization of the lymphatics using immunohistochemistry [4].

With burgeoning research, the ability of the lymphatic vasculature to modulate numerous pathological and physiological processes is now well-recognized [3]. A defective lymphatic system may lead to the accumulation of pro-inflammatory cells, signalling molecules and oedema, ultimately resulting in a feedback loop that fuels a state of chronic inflammation [5]. Indeed, the lymphatic system plays a critical role in immunity as a surveillance network running in parallel with the blood vasculature and providing continuous information to the immune system [1,5]. In particular, the lymphatics are a conduit for inflammatory cell clearance from the site of injury to the lymph nodes, thus enabling cross-talk of the innate and adaptive immune system [1,6]. Lymphatic impairment is correlated with the pathogenesis of multiple chronic inflammatory diseases, such as cardiovascular disease, inflammatory bowel disease (IBD) and dry eye disease (DED) [1,5,7,8].

In healthy individuals, upon acute tissue injury or host infection, an inflammatory cascade is triggered as a protective response by resident cells (e.g., mast cells, tissue macrophages, dendritic cells and/or endothelial cells) [9]. A storm of pro-inflammatory soluble mediators is produced and released that activate the innate immune response. Upregulated expression of integrins and adhesion molecules on local blood vessel endothelial cells bind circulating leukocytes, resulting in an influx of polymorphonuclear cells (PMNs) from the blood circulation [9]. PMNs kill and clear foreign invaders and/or damaged cells through multiple mechanisms, including phagocytosis, release of proteases, free radicals and/or neutrophil extracellular traps [10]. More leukocytes are recruited depending on the inflammatory stimulus, such as T-cells and monocytes that differentiate into macrophages [11]. During the peak of inflammation, as the inflammatory stimulus is being eliminated, the resolution phase begins. Clearance of inflammatory stimuli leads to a reduction of pro-inflammatory signals and a switch from the production and release of pro-inflammatory lipid mediators to specialized pro-resolving mediators (SPMs) [12]. The production and release of SPMs and anti-inflammatory cytokines halt PMN infiltration, while stimulating the recruitment and function of monocytes and pro-resolving macrophages [9]. The pro-resolving macrophages clear the tissue of cellular debris in a non-phlogistic manner [13]. Leukocytes are cleared by either undergoing apoptosis and subsequently being efferocytosed by other phagocytes, and/or by egressing the tissue through the lymphatics [9,12]. Resolution of inflammation is achieved by complex coordinated signals that overlap each other and ultimately result in tissue homeostasis. 

Many chronic diseases are characterized by impaired resolution and a vicious cycle of continuous immune cell activation [14,15,16]. Impaired lymphatic function and abnormal morphology is found in conjunction with states of chronic inflammation [1,17,18,19,20]. This review provides an overview of the abundance, and the effects of SPMs on the lymphatic system in relation to inflammatory diseases such as cardiovascular disease, acute respiratory distress syndrome, IBD and DED.

## 2. The Lymphatic System

The lymphatic system consists of lymphoid organs and a unidirectional vascular network spanning throughout the body [1,20,21]. The lymphatic vasculature is intertwined through almost all tissues and organs except for bone marrow, cartilage, retina, lens, brain and epidermis [3,20]. The initial lymphatic capillaries in peripheral tissues are blunt-ended vessels consisting of a single layer of lymphatic endothelial cells (LECs). The single layer of LECs has “button-like” junctions that act as an opening to allow particles to enter the lymphatic lumen due to a pressure gradient [1,3,20]. Active mechanisms are also involved in the transportation of particles into the capillaries and to the lymph node. For example, the lacteals, which are lymphatic capillaries in the small intestine, interact with enterocytes to actively transport lipids [22,23]. In addition, immune cells express various integrins that bind to LECs to enable their migration through to the lumen of the lymphatic vessel, where they are subsequently drained to the lymph node [24]. The initial capillaries join into larger afferent collecting lymphatic vessels that are lined with smooth muscle cells pumping lymph towards the draining lymph node [1,20]. The collecting lymphatic vessels have “zipper-like” junctions which are tighter and surrounded by a continuous basement membrane with intraluminal valves that prevent backflow [3]. Interstitial fluid, macromolecules and immune cells are transported from the periphery to the draining lymph node, and eventually through the thoracic duct or right lymphatic trunk and back into venous blood circulation [3].

In the 1990s, lymphatic research flourished with the ability to visualize LECs. Key markers of LECs are lymphatic vessel endothelial hyaluronan receptor-1 (LYVE-1) [25], integral membrane glycoprotein podoplanin (PDPN) [26], prospero homeobox-1 (PROX-1) [27] and vascular endothelial growth factor receptor-3 (VEGFR-3) [28]. It is important to note that the use of a combination of these markers is necessary for distinguishing LECs from blood endothelial cells, as they are not uniquely expressed on respective cell types. For example, even though PROX-1 is thought to be a master-regulator of endothelial cell lymphatic phenotype, it is also expressed on cardiac valves and venous valves [3]. Furthermore, VEGFR-3 is expressed on both blood and lymphatic vasculature during early development, but its expression declines on blood endothelial cells during the stage of lymphatic budding [4,29,30]. VEGFR-3 expression can continue to be found into adulthood on fenestrated vessels or high endothelial venules [4,31]. These examples highlight the importance of utilizing a minimum of two markers to identify LECs.

In adulthood, lymphangiogenesis is only initiated during inflammation, tissue repair or tumour-related conditions, and dysregulation is observed in several pathological conditions [32,33,34,35]. VEGFR-3 is strongly expressed by adult LECs and one of its ligands, vascular endothelial growth factor (VEGF)-C, is the most well-characterized signalling mechanism for inducing lymphangiogenesis [1]. In VEGF-C^−/−^ mouse embryos, endothelial cells expressing PROX-1 (that are committed to the lymphatic phenotype) are unable to proliferate from the cardinal vein during development resulting in no live-born pups due to an impairment of the lymphatic vasculature [36]. In addition, overexpression of VEGF-C leads to significantly higher lymphatic drainage due to an increase in size and abundance of lymphatic vessels [21]. Thus, lymphangiogenesis is important for drainage and clearance at the sites of inflammation and blockage causes impaired lymphatic function, which exacerbates the inflammatory response [21,34,37].

The lymphatic system plays a critical role in maintaining tissue homeostasis and resolving inflammation by balancing interstitial levels of fluid. In inflamed tissue, leaky inflammatory blood vessels contribute to oedema that increases the interstitial pressure. This increase in pressure dilates the lymphatic capillaries and enables the clearance of residual tissue fluids, thus resulting in homeostasis [1]. Interestingly, although the human body contains three litres of plasma, it is estimated that approximately eight litres of fluid is extravasated systemically daily due to hydraulic pressure and approximately half is taken up by the lymphatics through colloid osmotic pressure [38]. 

Additionally, the lymphatics act as a channel of communication between the site of tissue injury and the lymph nodes, by transporting signalling molecules, exosomes, soluble antigens and antigen presenting cells to the lymph node, as a crucial part of the initiation of adaptive immune responses [6,39,40,41,42,43]. It is thought that soluble antigens are rapidly transported to the draining lymph node to prime resident immune cells for the coming antigen presenting cells [40,44,45]. Furthermore, LECs control immune cell migration from tissue to the lymph node through adhesion molecule expression and chemokine production [3]. The most explored signalling axis for immune cell migration is the CCR7 receptor (expressed on immune cells) and its ligands CCL21 and CCL19 (produced by LECs into the lymphatic lumen). For example, CCL21 and CCL19 are potent chemokines responsible for dendritic cell and T-cell migration to the draining lymph nodes [46,47]. Interestingly, regulatory T-cells modulate LEC adhesion molecules, which enables other leukocytes to cross over into the lymphatic lumen [48]. During the resolution of inflammation, after efferocytosis (i.e., removal of apoptotic cells by phagocytes) has occurred, the phagocytic cells may be cleared from the site of inflammation by migrating towards the chemotactic gradient into the lymphatic lumen, towards the lymph node and potentially back into circulation via the afferent lymph [43,49]. The lymphatics thus provide a conduit of communication between the site of inflammation and the adaptive immune system.

Although the brain lacks lymphatic vessels, it was recently discovered that cerebrospinal fluid and interstitial fluid from the brain parenchyma are drained through meningeal lymphatic vessels to the cervical lymph nodes [50]. Lymphatic vessels are found in the dural layer of the meninges at the dorsal and ventral side of the brain, as well as at the cribriform plate, where the olfactory neurons cross the ethmoid skull bone [51,52]. In the experimental autoimmune encephalomyelitis (EAE) model of the inflammatory condition, multiple sclerosis, there is expansion of the lymphatics at the cribriform plate that leads to increased communication between the brain and the cervical lymph nodes and perpetuates the neuroinflammatory state [53]. This suggests that there is an effective passage of antigens from the brain to the cervical lymph nodes with a feedback loop that stimulates inflammatory responses in the central nervous system, which could possibly be targeted by pro-resolving mediators. Furthermore, meningeal lymphatic structures may affect post-stroke outcomes [54]. Interestingly, recent evidence suggests that SPMs may have potential as novel anti-stroke therapy [55] and it would be important to elucidate if this outcome is interlinked with meningeal lymphatic function.

## 3. Specialized Pro-Resolving Mediators in Resolution of Inflammation

The medical community has begun to target chronic inflammation through pro-resolution strategies, in addition to the more well established anti-inflammatory therapeutics [56]. Indeed, the discovery of the first potent SPMs [57] has revolutionized the potential of treatment for chronic inflammation [9,14,58,59]. SPMs are a class of signalling bioactive lipid metabolites produced endogenously to promote resolution of inflammation. They are derived from omega-3 fatty acids (eicosapentaenoic acid (EPA) and docosahexanoic acid (DHA)) or omega-6 fatty acids (arachidonic acid (AA)) in a cyclooxygenase (COX) and lipoxygenase (LOX) dependent manner [59].

AA-derived Lipoxin A_4_ (LXA_4_) and Lipoxin B_4_ (LXB_4_) produced by human leukocytes were the first SPMs discovered in 1984 [60]. There are different pathways of lipoxin production that involve transcellular biosynthesis during cell cross-talk in inflammation [61,62,63]. One biosynthetic route is lipoxygenation (insertion of molecular oxygen) of AA by 15-LOX in monocytes, airway epithelial cells or eosinophils, which generates 15*S*-hydroperoxyeicosatetraenoic acid (15*S*-H(p)ETE) or 15*S*-hydroxyeicosatetraenoic acid (15*S*-HETE). These lipoxin precursors are subsequently substrates for 5-LOX in neutrophils (Figure 1). Notably, leukotriene synthesis is halted by 5-LOX activation, and thus lipoxin generation can also cause decreased production of the pro-inflammatory eicosanoids [62,63]. Another lipoxin-generating pathway involves an interaction between platelets and leukocytes. Upon neutrophil-platelet adhesion, the neutrophil produced AA-derived leukotriene A4 is converted to lipoxins by 12-LOX in the activated platelet (Figure 1) [61]. Yet another mechanism of lipoxin synthesis is in neutrophils through the esterification of 15-HETE in inositol-containing phospholipids [64]. Of note, lipoxins may also be generated from individual cells’ endogenous AA, if they are primed by an inflammatory state [62]. Aspirin can elicit the production of so-called aspirin triggered lipoxins (ATLs), aka 15-epi-LXs, which have increased stability and potency [65]. Briefly, low-dose aspirin irreversibly acetylates COX-2 that then generates 15*R*-HETE and subsequently 15-epi-LXs [62]. 

Lipoxins play multiple roles in the resolution of inflammation through the modulation of numerous immune cells [62]. For example, lipoxins regulate neutrophils by inhibiting recruitment to sites of infection, production of superoxide anion and degranulation [66]. Lipoxins increase monocyte chemotaxis and promote a pro-resolving macrophage phenotype, thus enhancing efferocytosis [59,67]. They also inhibit production and release of multiple pro-inflammatory cytokines such as interleukin (IL)-12 from dendritic cells and/or IL-6 and IL-8 from epithelial cells, while stimulating anti-inflammatory cytokines such as IL-10 and Transforming growth factor (TGF)-β1, which also promote efferocytosis [14,59]. In addition, in mouse models of obesity, LXA_4_ treatment rescued high fat diet-induced decrease of key autophagy markers, P62 and LC-3 in adipocytes [68]. Lipoxins play a crucial role in resolution of inflammation and present a therapeutic strategy for combating chronic inflammation.

SPMs derived from EPA are the E-series resolvins, Resolvin E1 (RvE1), E2 (RvE2) and E3 (RvE3). E-series resolvins are produced in a similar manner to ATLs, through aspirin-acetylated COX-2, EPA transformation and subsequent metabolite conversion by 5-LOX in leukocytes [59,69,70]. E-series resolvins also prevent transmigration of neutrophils across both endothelial and epithelial cells [71,72], stimulate macrophage phagocytosis of apoptotic neutrophils [73], inhibit dendritic cell production of IL-12 and migration, among multiple additional immune regulatory activities [59,74]. 

Finally, SPMs derived from DHA are D-series resolvins, protectins and maresins [59]. There are D-series resolvins (RvD1 to RvD6 along with AT-RvD1 to AT-RvD6), two protectins (PD1 or AT-PD1) and two maresins (Mar1 and Mar2). DHA is transformed to 17(*S*)-hydroperoxyDHA (17(*S*)-HpDHA) by 15-LOX, that is then further metabolized by 5-LOX into D-series resolvins, maresins or protectins. DHA is converted by acetylated COX-2 into a metabolite substrate for 5-LOX to produce AT-resolvins. The pathways and structures of these active metabolites are described by Serhan et al. in great detail [75]. 

DHA-derived SPMs have a potent immunomodulatory effect on leukocytes as well. For example, RvD1, RvD3 and PD1 prevent neutrophil migration, while strongly promoting macrophage phagocytosis and efferocytosis [14,59,76]. RvD1 inhibits lipopolysaccharide-induced pro-inflammatory tumour necrosis factor (TNF)-α secretion from macrophages, while PD1 also inhibits TNF-α and interferon (IFN)-γ secretion by neutrophils [77]. PD1 also reduces angiogenesis and promote epithelial barrier integrity [9,78].

SPMs have a critical role in modulating various immune cells during the resolution of inflammation to promote tissue repair and homeostasis [79]. Thus, SPMs may have significant therapeutic potential in the treatment of chronic inflammatory diseases by promoting resolution through the human body’s endogenous pathways.

## 4. Lymphatics in Inflammatory Diseases

### 4.1. Cardiovascular Disease

Atherosclerosis is a major cause of cardiovascular disease, e.g., heart failure, stroke and myocardial infarction (MI). Atherosclerosis develops over decades and is characterized by the build-up of cholesterol and immune cells within lesions of the arterial wall of medium or large arteries. Endothelial dysfunction results in accumulation of apolipoprotein B (apoB) and low-density lipoprotein (LDL) cholesterol in the arterial wall. Monocytes are recruited into the subendothelial space and phagocytose apoB containing lipoproteins, leading to the formation of foam cells. Heightened inflammation results in an increase of oxidative stress, which upregulates integrin expression on blood endothelial cells, resulting in additional monocyte recruitment [80]. The continuous recruitment of monocyte-derived macrophages result in an accumulation and build-up of macrophages and foam cells in the plaque, which contribute to disease progression [81]. Eventually the core of the plaque becomes necrotic and along the endothelial lining of the plaque, a fibrous cap of smooth muscle cells and collagen develops [82]. The plaque transitions from preclinical to clinical when it has grown so large it disturbs the blood flow [83]. Atherosclerosis usually goes unrecognized until the plaque ruptures, which is one of the most direct causes of cardiovascular disease.

Regression of atherosclerosis occurs with cholesterol removal from foam cells, known as reverse cholesterol transport (RCT) [84]. Interestingly, lymphatic vessels are a route for cholesterol clearance from tissue and plaques [85]. For example, in two models of impaired skin lymphatic drainage, injection of cholesterol loaded-macrophages resulted in their accumulation upstream of the damaged lymphatic vessels [81], indicating that lymphatic function is necessary for cholesterol transport. Furthermore, Apolipoprotein E (*apoE*) deficient mice treated with anti-VEGFR-3 antibody received an atherosclerotic aortae loaded with labelled cholesterol transplant. The cholesterol remained in the aortae of the anti-VEGFR-3 antibody treated mice, demonstrating the critical role of the lymphatic vasculature in RCT [81]. Additionally, in an atherosclerotic mouse model, lymphangiogenesis induced by VEGF-C led to increased RCT and thus decreased cholesterol content in the plaque [85]. Moreover, disruption of collecting lymphatic vessels decrease RCT by 80% [85]. Conversely, stimulating lymphangiogenesis by treating plaque prone mice (LDL receptor knockout on a high fat diet) with VEGF-C led to improved lymphatic molecular transport, reduced plaque formation, limited macrophage accumulation and enhanced immune cell migration through the lymphatics, as compared to the control group [86]. These results demonstrate the importance of lymphatics for cholesterol clearance and thus reduction of atherosclerosis.

The lymphatic vasculature is not only critical in preventing atherosclerotic plaques, but also in recovery post MI [87,88]. Post MI, VEGF-C induced lymphangiogenesis improves clearance of immune cells into the draining mediastinal lymph nodes in a LYVE-1 dependent manner. Inhibiting LYVE-1 docking resulted in intensified chronic inflammation and a decline in cardiac function [87]. Blocking VEGFR-3 signalling resulted in a more intense MI and significantly higher rate of mortality [89]. Thus, in summary, the lymphatics play a critical role in both the prevention of cardiovascular diseases, and the recovery after myocardial events (see most recent review [90]), where an important function is promoting resolution of inflammation.

### 4.2. Inflammatory Bowel Disease

IBD (encompassing Crohn’s disease and ulcerative colitis) is thought to be a multi-factorial disease that includes a dysregulated immune response to the microbiome, whereby the gastrointestinal immunity is activated and remains in a chronic inflammatory state. In the gastrointestinal tract of patients with IBD, the lymphatics are abnormal, dilated, obstructed, have impaired function along with oedema and increased lymphangiogenesis [91,92]. IBD patients also have impaired immune cell migration, lymph drainage and lymphatic pumping [1]. Interestingly, surgical intervention that reduced the gastrointestinal inflammatory response also restores lymphatic morphology and function in patients with Crohn’s disease [93]. In another clinical study, Crohn’s recurrence was correlated with decreased density of lymphatic vessels [94]. Additionally, multiple animal studies have tested the role of lymphatics in attenuating disease using mouse models of IBD, such as IL-10-knockout mice or dextran sulphate sodium-induced colitis [95,96]. Adenoviral delivery of VEGF-C to these IBD mouse models resulted in an increase in lymphatic vessel density, augmented inflammatory immune cell migration and bacterial clearance, leading to an overall reduction in disease severity [97]. To further demonstrate the importance of the lymphatic vasculature in IBD, VEGFR-3 blockade in IL-10 deficient mice led to a significant increase in disease severity with increased inflammation, impaired lymphatic function and morphology along with oedema in the colon [98]. It is important to note that it is not yet known whether the changes in the lymphatics cause the inflammatory response, or whether the inflammation associated with IBD impair the lymphatics. Recent studies investigating impaired resolution of inflammation are correlating it with the onset of IBD [16]. SPMs can alter disease progression in IBD by slowing neutrophil transmigration across the intestinal vasculature, preventing release of inflammatory cytokines and promoting the production of other SPMs, thus they offer a potential novel mechanism to target IBD [16].

### 4.3. Dry Eye Disease

DED is an immune-regulated disorder that is characterized by ocular surface inflammation due to tear dysfunction [7]. The pathogenesis remains unknown but evidence of activation of stress signalling pathways by resident cells leads to the accumulation of inflammatory molecules and a subsequent feedback cycle of continuous inflammation [99]. Increasing evidence is pointing towards the role of T-cells in disease development [99]. In mice with DED, T-cells are activated in the lymph nodes and chemokines continue to recruit T-cells to the inflamed ocular surface [7]. The draining lymph nodes play a crucial role in acceptance of corneal transplantation. Removal of the draining lymph node in mice leads to increased survival rates of corneal transplants [100]. Thus, reducing inflammatory activation in the lymph node by promoting resolution represents a novel potential target to promoting graft acceptance in cases of corneal neovascularization. In addition, topical application of SPMs have been shown to reduce inflammation in DED [101].

## 5. Specialized Pro-Resolving Mediators and the Lymphatic System

Increasing evidence indicates that SPMs play a role in regulating resolution of inflammation in relation to the lymphatics and may be a novel therapeutic target for combating inflammatory disease (Table 1).

### 5.1. Specialized Pro-Resolving Mediators in the Lymph Nodes and Spleen

SPMs are naturally occurring in lymphoid organs, but their abundance and roles have yet to be fully elucidated. In human lymph nodes and spleens, distinct profiles of SPMs have been identified by liquid-chromatography-mass spectrometry (LC-MS/MS) [107]. Targeted LC-MS/MS allowed for SPM detection at ∼0.1 pg by adding labelled standards. In axillary lymph nodes (from patients with Stage IV breast cancer, cardiorespiratory failure or craniocerebral injuries) the D-series resolvins (RvD1, RvD5, and RvD6) along with the E-series (RvE3) were most prominent. In other lymph nodes, LXA_4_ and LXB_4_ were identified along with the pro-inflammatory LTB_4_ and 5S,12S-dihydroxy-6,8,10,14-(E,Z,E,Z)-icosatetraenoic acid (5S,12S-diHETE) from patients with Crohn’s disease, liver cirrhosis or heart failure. Prostanoids were also detected, with highest levels of PGE_2_, which can serve both pro- and anti-inflammatory roles (Table 2) [107,108]. Similarly, pro-inflammatory LTB_4_ and pro-resolving lipoxins are found within the same lymph nodes from patients with inflammatory diseases [107], which may be due to the switching of eicosanoid synthesis. The balance between pro-inflammatory mediators and pro-resolving mediators within the lymph nodes of patients with chronic inflammatory conditions is important to observe. As described in the detailed review by Dr. Serhan and Dr. Savill, “… the beginning programs the end,” as it is proposed that prostaglandins activate the transcription of enzymes responsible for lipoxin, resolvin and protectin synthesis [58]. Thus, in a chronic inflammatory state it is plausible that there is an ongoing “flickering of the switch” between pro-inflammatory and pro-resolving lipid mediator synthesis. 

In spleens from patients with sepsis and either leukaemia, heart failure or pneumonia, a variety of omega-3-derived and eicosanoid-derived lipid mediators were identified: RvD5, RvE1, RvE2, RvE3, PD1, 10S,17S-diHDHA, PGD_2_, PGE_2_, PGF2α, MaR2, 7S,14S-diHDHA and LXA_4_ (Table 2) [107]. Futures studies on SPMs in lymphoid organs from healthy individuals without chronic inflammatory diseases would be of interest to determine the basal state of SPM production in humans, although these are understandably challenging to obtain. Alternatively, various disease models could be used to compare healthy versus unhealthy SPM profiles of lymphoid organs to determine the SPM composition and elucidate their roles during recovery from inflammatory diseases.

The spleen is the largest lymphatic organ in the body and plays a critical role in cardiac remodelling post MI [104,109]. In a non-inflammatory state, neutrophils are present in the spleen near B-lymphocytes, and upon inflammation, their numbers increase. In a mouse model of MI induced by surgical ligation of the left anterior descending coronary artery, RvD1 enhanced recovery post-MI [104]. Specifically, RvD1 protected the splenic reservoir of immune cells, decreased oedema, halted neutrophil recruitment to the left ventricle and improved left ventricle function. RvD1 also altered the production of other SPMs in the spleen with a notable increase of RvD2, 17R–RvD1, MaR1, LXA_4_ and LXB_4_ [104], which likely accelerated the resolution. Additionally, RvD1 treatment resulted in a rapid clearance of macrophages [104], potentially through the lymphatics. The abundance of SPMs in lymphoid organs and their ability to accelerate post-MI recovery highlight their therapeutic potential.

**Table 2 ijms-22-02750-t002:** In vivo detection of specialized pro-resolving mediators in lymphatic system.

Species	SPMs	Location/Disease	Sample Collection Time	Refs
*Homo Sapiens*	RvD1, RvD5, RvD6 and RvE3	Axillary lymph nodes (from patients with Stage IV breast cancer, cardiorespiratory failure or craniocerebral injuries).	Post mortem	[107]
	PGE2, LXA_4_ and LXB_4_	Lymph nodes (from patients with Crohn’s disease, liver cirrhosis or heart failure).	Post mortem	
	RvD5, RvE1, RvE2, RvE3, PD1, 10S,17S-diHDHA, PGD2, PGE2, PGF2α, MaR2, 7S,14S-diHDHA and LXA_4_	Spleens (from patients with sepsis along with leukaemia, heart failure or pneumonia).	Post mortem	
African Green Monkey	LXA_4_ and LXB_4_, DHA-derived 7S,14S-diHDHA	Terminal ileum in Crohn’s disease model. Surgically induced lymphatic obstruction.	7 days post-injury	[110]
	D-series resolvins, protectins, and maresins		61 days post-injury (significantly increased than day 7)	
	MaR1, 7S,14S-diHDHA, RvD5, RvD6, PDX, AT LXA_4_, LXB_4_, LTB4, and RvE2	Colon in Crohn’s disease model. Surgically induced lymphatic obstruction.	21 days post-injury	

### 5.2. Specialized Pro-Resolving Mediators in the Gastrointestinal Tract

Crohn’s is an autoimmune chronic inflammatory condition of the gastrointestinal tract [16], which is an area where SPMs have been proposed as novel form of treatment to enhance inflammatory resolution and promote disease remission [111,112]. As described above, IBD and Crohn’s result in impaired lymphatics. Indeed, Crohn’s disease can be modelled by surgically obstructing the lymphatics in the African green monkey. This model was used to measure levels of SPMs in the gastrointestinal tract in response to injury (Table 2) [110]. The peak of the inflammatory response, recorded 7 days post lymphatic obstruction, was characterized by the increase in inflammatory cytokines and chemokines. At this time, a lipid mediator class switch towards SPM production was already observed in the terminal ileum, as evidenced by an increase in the pro-resolving AA-derived LXA_4_ and LXB_4_, DHA-derived 7S,14S-diHDHA and a decrease of AA-derived pro-inflammatory PGE_2_ and LTB_4_ [110]. Interestingly, the SPM production kept increasing until day 61 post-injury [110], which may be the “post-resolution phase”. The post-resolution phase is often overlooked and was initially described by Newson et al. [113]. They speculated that this phase consists of a second influx of leukocytes, with monocyte-derived macrophages and T-cells remaining in the tissue for months to regulate additional inflammatory responses, and that resolution in this way acts as a bridge between innate and adaptive immunity, as well as tissue reprogramming [113]. It would be interesting to visualize changes in lymphatic density and morphology prior to and after the post-resolution phase, as macrophages are important inducers of lymphangiogenesis. 

Importantly, unlike the small intestine, the colon had a global reduction in all lipid mediators at day 61 post-injury, with maximum levels of SPMs at day 21 post-injury, indicating the tissue specificity of the pro-resolving response [110]. 

The explorative study by Becker et al. [110] that investigates the levels of SPMs post lymphatic injury is of interest as it provides insight into how/when SPM treatment could be used to enhance resolution and reduce recovery time. Additionally, others have experimented with different timelines of treatment with SPMs in mouse models, as described in the section below.

### 5.3. Specialized Pro-Resolving Mediators in the Lungs

Changes in lung lymphatic vasculature are observed in numerous diseases [102,114,115,116], where preventing pulmonary oedema is of paramount important to facilitate breathing. Treatment with AT-RvD3 and RvD3 has been explored in a mouse model of acute respiratory distress syndrome induced by injection of hydrochloric acid into the mainstem bronchus [102]. Mice treated with AT-RvD3 1 h post lung injury demonstrated decreased correlates indicating acid injury, including oedema, alveolar wall thickening, haemorrhage and leukocyte influx [102]. Furthermore, epithelial sodium channel (ENaC)γ expression, an indicator of alveolar water clearance, was significantly increased in the lungs of mice treated with AT-RvD3 [102]. These results combined indicate that AT-RvD3 enhances resolution of oedema, i.e., fluid clearance through the lymphatics. Additionally, AT-RvD3 treatment led to a significant increase in LYVE-1+ vessels and a slight increase in VEGFR3+ vessels 72 h post injury. These results demonstrate that SPMs play a role in inducing lymphangiogenesis in the lung. A notable reduction in neutrophil numbers was also observed at 6, 24, and 72 h post AT-RvD3 treatment [102]. The authors noted that this was due to a reduction in neutrophil infiltration, presumably due to SPMs actions as these lipids are known halt neutrophil recruitment [102]. However, one may speculate that the continuous significant decrease in neutrophils may also be due to clearance through the lymphatics. Future studies with labelled leukocytes would be of interest to track whether AT-RvD3 is solely inhibiting neutrophil influx, or if it is also promoting clearance through the lymphatics to the lymph nodes and/or the spleen. In summary, this study shows that AT-RvD3 has the potential to treat acute respiratory distress syndrome by promoting resolution and modulating the lymphatics [102]. Further studies measuring the ability of AT-RvD3 treatment to promote lymphangiogenesis after full recovery would be of interest, as the following study described below found little effect using other SPMs.

### 5.4. Specialized Pro-Resolving Mediators and the Eyes

Angiogenesis has been linked to exacerbated inflammation, while lymphangiogenesis has largely been shown to be beneficial in resolving inflammation. The effects of AT-LXA_4_, RvD1 or RvE1 on angiogenesis and lymphangiogenesis were assessed in a model of corneal neovascularization induced in mice by suture or micropellet (containing IL-1β and VEGF-A). AT-LXA_4_, RvD1 or RvE1 treatment significantly inhibited angiogenesis, but in this model the SPMs had little to no effect on lymphangiogenesis or expression of VEGF-D and VEGFR-3 [103]. All treatments did however lead to a significant reduction of neutrophils 24 and 72 h and of macrophages 72 h post injury at the suture site [103]. Although the authors suggested that this is due to reduced infiltration, one may speculate that it could also be due to increased egression through the lymphatics. It would thus be interesting to measure labelled neutrophil and macrophage counts in the draining lymph node to determine if these treatments can promote immune cell egression through the lymphatics as mentioned above.

Interestingly, dietary DHA supplementation improved outcomes of immune-driven DED in a mouse model. An established model of desiccating stress to mimic DED using a combination of continuous airflow, low humidity and inhibiting lacrimal gland function was used to assess the effects of a diet enriched (40%) in DHA [117]. Remarkably, dietary DHA supplementation resulted in a significant increase of LXA_4_ levels in the draining lymph nodes [105]. Notably, 17-HDHA and 14-HDHA, DHA-SPM pathway intermediates, were also significantly increased in draining lymph nodes, along with in the corneas and lacrimal glands. The reason that DHA would increase an AA-derived may be due to the feed-forward mechanism whereby SPM treatment increases production of other SPMs [118]. DHA-supplementation also resulted in a significant increase in LXA_4_ production from lymph node-derived regulatory PMNs. On the contrary, PMNs from DHA-deficient mice produced a significantly higher amount of the pro-inflammatory LTB_4_. In DED, lymph node PMNs were increased overall, but DHA supplementation only increased PMN levels in the draining lymph node in female but not male mice [105]. Acute-DHA supplementation was able to rescue PMN lymph node numbers in DHA-deficient mice and increase LXA_4_ levels [105]. Thus, DHA dietary supplementation may be a therapeutic strategy to target SPM production in the lymph nodes and potentially reduce corneal transplant rejection by decreasing the inflammatory response. This research demonstrates the potential of DHA-supplements to regulate lymph node immune cells through the SPMs.

### 5.5. Specialized Pro-Resolving Mediators and the Peritoneum

To study the interaction of immune cells and the lymphatics, multiple studies have used zymosan-induced peritonitis. In this model, eosinophils have a particularly important role as major players both in the acute phase of inflammation, but also during resolution when they release important 12/15-LOX pre-cursors. It is noteworthy that depletion of eosinophils resulted in impaired lymphatic drainage in murine zymosan-induced peritonitis [106]. In eosinophil-depleted mice, the impaired lymphatic drainage was rescued by intraperitoneal injection of PD1, demonstrating that the impaired drainage was due to a lack of PD1 production at the site of inflammation by eosinophils [106]. In addition, eosinophils promote immune cell clearance through regulation of CXCL13 by a 12/15-LOX-dependent mechanism. CXCL13 depletion resulted in decreased phagocyte drainage to lymph nodes. In eosinophil deficient mice, LXA_4_ administration increased CXCL13 expression and rescued the impaired lymph node hypertrophy [72]. This potential for PD1 and LXA_4_ to rescue lymphatic function and morphology should be investigated further. Future experiments should continue to explore the effects of various SPMs on the lymphatic drainage and lymph node morphology in models of various chronic inflammatory diseases.

Further, SPMs demonstrate the ability to modulate immune cell infiltration and migration across the blood endothelium as well as the lymphatic endothelium. In zymosan-induced peritonitis, PD1 injected intraperitoneally reduced PMN infiltration by over 40% [49]. ATLs reduced PMN influx by 26% and RvE1 increased mononuclear cell infiltration by approximately 24% from the circulation. On the resolving side of inflammation, treatment with ATLs 24 h after inflammatory insult resulted in approximately double the amount of zymosan-carrying leukocytes in the draining lymph nodes and along with a significant increase in the spleen, indicating enhanced immune cell migration. Furthermore, RvE1 also significantly increased zymosan-carrying leukocytes in the spleen, but with only a slight increase in the lymph node. However, PD1 had the most impressive effects on clearing zymosan-carrying leukocytes through the lymphatics with PD1 treatment increasing the numbers more than four-fold in the spleen and more than six-fold in the lymph nodes [49]. Schwab et al. further demonstrated the importance of SPMs in promoting leukocyte egression through the lymphatics by inhibiting COX-2 and LOX. COX-2 inhibition resulted in > 50% decrease in the amount of zymosan in the lymph nodes and the spleen, which was rescued by treatment with either AT-LXA_4_, RvE1 or PD1_._ Similar results were found with LOX inhibition to a slightly lesser extent [49]. Additional studies have shown that blocking SPM biosynthesis with either LOX or COX inhibitors resulted in leukocyte accumulation at the site of the inflammation because of impaired phagocyte removal to the lymphatics [49,106,119]. Immune cell migration from sites of inflammation to draining lymph nodes or the spleen is necessary to resolve inflammation. The impressive scale to which these SPMs can promote immune cell clearance would be extremely useful in treating diseases where accumulation of inflammatory immune cells creates a vicious loop of continuous leukocyte recruitment.

## 6. Conclusions

Impaired lymphatic function and abnormal morphology are characteristic features of diseases with a chronic inflammatory component. Indeed, the lymphatic system plays an important role in the body’s ability to resolve an inflammatory insult, as it provides a conduit for fluid, immune cells, macromolecules, and a route of communication between the innate and adaptive immune system. Thus, restoring dysregulation of the lymphatic system may be a promising therapeutic target for breaking the pro-inflammatory feedback loop within chronic diseases. 

Experimental models indicate that there is a possibility to target chronic inflammation by promoting resolution through therapeutic application of SPMs, in order to enhance lymphatic function and attenuate disease. Treatment with supplementing bioactive lipid mediators has the potential to enhance the body’s endogenous resolution pathways to prevent excessive inflammation and promote tissue remodelling. Specifically, SPMs induce resolution of inflammation by promoting leukocyte egression from the site of tissue injury through the lymphatics, decreasing oedema, stimulating lymphangiogenesis and repairing lymph node hypertrophy. SPMs are readily detectible in human lymphoid organs and studies in primates show that their relative abundance is altered during the different stages of healing after lymphatic obstruction. This provides insight into the body’s endogenous resolution pathways in lymphatic disease. 

In summary, the current literature provides support for continued investigation of SPMs as a potential therapeutic approach to rescue lymphatic function in diseases characterized by chronic inflammation.

## Figures and Tables

**Figure 1 ijms-22-02750-f001:**
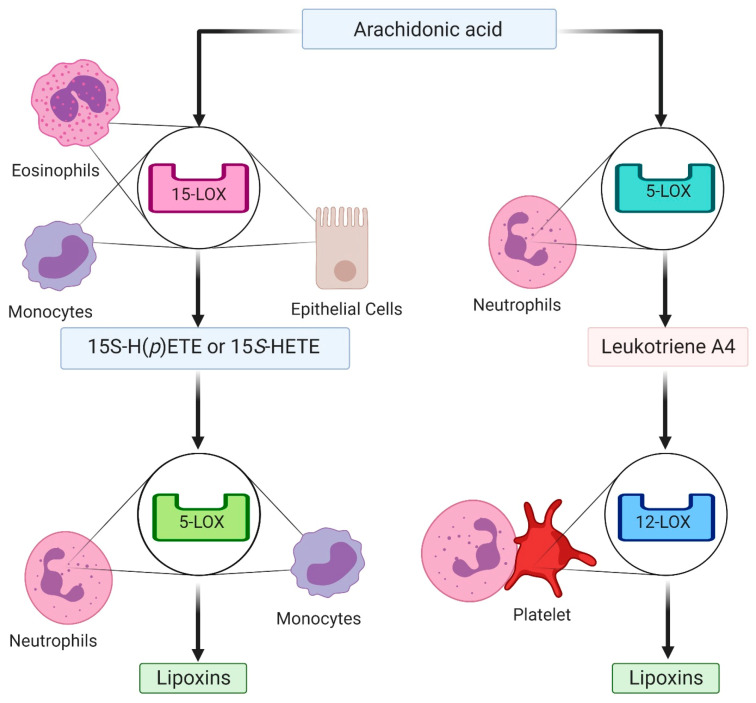
Arachidonic acid-derived specialized pro-resolving mediator synthesis. Synthesis of the specialized pro-resolving mediator, lipoxin, requires transcellular biosynthesis (left side). Arachidonic acid receives an oxygen from 15-lipoxygenase (LOX), generating either 15*S*-hydroperoxyeicosatetraenoic acid (15*S*-H(*p*)ETE) or 15*S*-hydroxyeicosatetraenoic acid (15*S*-HETE). These metabolites are taken up by neutrophils where 5-LOX generates 5,6-epoxytetraene, an unstable molecule which is hydrolysed into lipoxins. Arachidonic acid can also be metabolized by neutrophils into leukotriene A4 (right side). Upon neutrophil-platelet binding, leukotriene A4 is converted into lipoxins by platelet 12-LOX.

**Table 1 ijms-22-02750-t001:** Role of specialized pro-resolving mediators in lymphatic system.

SPMs	Disease Model	Species	Molecular Effect	Refs
AT-RvD3 and RvD3	Acute respiratory distress syndrome	Mouse—injection of hydrochloric acid into the mainstem bronchus	Decreased oedema, alveolar wall thickening, haemorrhage and leukocyte influx	[102]
			Significantly increased LYVE-1+ vessels and increase VEGFR3+ vessels post injury	
AT-LXA_4_	Corneal neovascularization	Mouse—suture or micropellet (containing IL-1β & VEGF-A)	Inhibited angiogenesis but not lymphangiogenesis. No effect on lymphangiogenic VEGF-D or VEGFR-3, reduction of neutrophils post injury	[103]
	Peritonitis	Mouse—zymosan induced	Reduced PMN infiltration. Doubled the amount of zymosan-carrying leukocytes in the draining lymph nodes and a significant increase in the spleen	[49]
RvD1	Corneal neovascularization	Mouse—suture or micropellet (containing IL-1β & VEGF-A)	No effect on lymphangiogenesis or expression of lymphangiogenic VEGF-Dor VEGFR-3. Decreased neutrophil count post injury	[103]
	Post-myocardial infarction	Mouse—surgical ligation of the left anterior descending coronary artery	Conserved transient changes in splenic weight following MI. Reduced pulmonary oedema. Significant increase neutrophils present in spleen and rapid clearance. Altered production of SPMs in spleen. Induced rapid clearance of macrophages	[104]
RvE1	Corneal neovascularization	Mouse—suture or micropellet (containing IL-1β & VEGF-A)	Prevented angiogenesis. Did not alter lymphangiogenesis or VEGF-D or VEGFR-3 expression. Reduced neutrophils post-injury.	[103]
	Peritonitis	Mouse—Zymosan induced	Increased mononuclear cell infiltration, Increased zymosan-carrying leukocytes in the spleen	[49]
DHA supplement (diet enriched for 40% DHA)	Dry eye disease	Mouse—desiccating stress (combination of continuous airflow, low humidity and inhibiting lacrimal gland function)	Increased LXA_4_ levels in the draining lymph nodes, increased LXA_4_ production from lymph node-derived regulatory PMN	[105]
PD1	Peritonitis	Mouse—zymosan induced	Rescued impaired lymphatic drainage in eosinophil-depleted mice. Reduced PMN infiltration. Increased zymosan-carrying leukocyte to the spleen 4-fold and 6-fold in lymph nodes	[106]
LXA_4_	N/A	Mouse—eosinophil-depleted mice	Rescue impaired lymph node hypertrophy	[72]

## Data Availability

Not applicable.

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
