# Peer review of "Specialized Pro-Resolving Mediators and the Lymphatic System"

_ijms, 2021, doi:10.3390/ijms22052750_

Round 1

Reviewer 1 Report

This review highlights previously underappreciated roles for SPMs in stimulating resolution of inflammation through the lymphatic system.

The manuscript as a whole has some great writing, but it could be shortened and detailed to really outline the nature of the work. More importantly, the major weakness of the article is that few papers clearly identify a connection between SPMs and the lymphatic system, since most of the discussed literature just speculate about this link. However, this innovative line of investigation could provide innovative approaches to target chronic inflammatory diseases and stimulating further research to rescue lymphatic function through SPMs is of paramount importance. Therefore, maybe a perspective article is more appropriate than a review.

In addition, the conclusion is not supported by the reported evidence. As it stands, the manuscript only “suggests” the investigation of SPMs as a potential therapeutic approach to rescue lymphatic function in diseases characterized by chronic inflammation.

In view of these considerations, I recommend working on the outline of the article to address these limitations.

Other points to be addressed:

  • Change “inflammatory resolution” with “resolution of inflammation” throughout the text
  • Abstract: the alteration of autophagy by SPMs is stated here but not described in the main text
  • Table 1 should be moved at the end of the paragraph since it covers also
  • Lines 217-220: Please, revise these sentences, they are difficult to understand
  • Paragraph 5.2: the description of the role of SPMs in modulating resolution of inflammation through the lymphatic is missing here. Please, clarify this point or remove this section.
  • Lines 407-418: no effect of SPMs in lymphatic.
  • Line 497-498: “mimic” is not appropriate since SPMs “are” endogenously produced bioactive molecules that stimulate resolution of inflammation

Author Response

We thank the reviewer for his insights and appreciate the suggestion to change the type of publication to a perspective article. We largely agree with the reviewer that a limited amount of work has been done that highlights links between SPMs and the lymphatics. However, in our view a review article is not only aimed at providing an overview on the latest progress in the field, but also points towards existing gaps, novel developments and directions the research field moves to and offers new insights based on the existing work of literature.  As such, we strongly believe that our article meets the criteria of a review article.

We agree with the reviewer and rephrased the conclusion to include that the evidence “suggests” that SPMs may serve as a potentially novel therapeutic approach, also highlighting that more work is needed to support this claim.

Reviewer 2 Report

This is interesting review on the specialized pro-resolving mediators (SPMs) and their role in the lymphatic system. Authors well balanced available data on the function of lymphatic system during inflammation and resolution and combined with current knowledge on the lipid mediators involved in these processes. I have only minor comments, which Authors could consider for corrections:

Line 69 radicals > free readicals

Line 70 into macrophades or T-cells

Line 129 the 8 L value is imprecise, at what time, what tissue. Daily flow at the thoracic duct is about 2,8 L, it is estimated 80% of total lymph turnover

Lines 299-303 I wonder if the model of corneal transplant is adequate for discussed dry eye disease. In the cited study [79] experiments were accomplished to exclude active immunotolerance by splenocytes adoptive transfer, and to support submandibular lymph node function in sensitization to allogenic corneal transplant and its rejection. Vascularisation of the recipient animal cornea was just a model to increase the risk of translant rejection (‘high-risk’ model). Perhaps dr Serhan’s studies on AT-lipoxins in preventions of corneal irritation would be better example in this manuscript.

Table 2 and Line 369 – Authors could introduce the acronym PDX or use proper protectin D1

Line 387 – epithelial sodium channel subunit abbreviation shoud be explained

Line 400 - > to treat

Author Response

We thank the reviewer for the valuable feedback, we have included all his/her suggestions and the valuable addition of studies on SPMs in DED.

If there are any further questions or suggestions, we would be happy to address them.

Reviewer 3 Report

The manuscript by Jamie D. Kraft et al. aims to provide an overview of the role of lymphatics in the resolution of inflammation and the interactions of pro-resolving mediators (SPMs). The topic is relevant and important. There is a great need for having a good review article discussing the latest developments of the field. The manuscript is well structured and easy to follow. However, I would suggest an accurate proofreading to correct all the typos and other grammatical problems.

The manuscript provides a good overview of the role of SPMs in inflammatory resolution. However, I would suggest the following revisions.

- Meningeal lymphatic vessels have been shown to govern inflammatory processes in the CNS and they are assumed to affect the pathophysiology of multiple sclerosis. Moreover, a recent article suggested that meningeal lymphatic structures may affect post-stroke outcomes and other findings propose the potential of SPMs in anti-stroke therapy. Therefore, it would be useful to provide a brief discussion of the connection of SPMs and meningeal lymphatics.

- Chapter 4 seems unnecessary to me. The cardiovascular, inflammatory bowel disease and dry eye disease related functions of lymphatics are discussed here without mentioning SPMs, and the role of SPMs is also summarized in other organs, moreover in a different context in chapter 5.

- In addition, I think the manuscript should be more focused, especially the introduction and chapter 2.

Author Response

We thank the review for his/her comments and appreciate the suggestions. Please find point-by-point answers to all comments below. We hope that the alterations render the manuscript now acceptable for publication in the eyes of this reviewer.

  1. We agree with the reviewer on the importance of meningeal lymphatics in governing inflammatory processes in the CNS, and the potential role of SPMs in anti-stroke therapy. We appreciate the suggestion to summarize recent findings in a brief discussion, and included a new paragraph at the end of section “2. The Lymphatic System”.
  2. We thank the reviewer for the comment on Chapter 4. We decided to keep this Chapter in the review, as it was meant to summarize the role of the lymphatics in inflammatory diseases for interested readers. However, we agree with the reviewer that the role of SPMs has not been discussed in this Chapter. We reworked the text to extend the emerging role that SPMs play for lymphatics-mediated resolution of these diseases.
  3. We appreciate the reviewer’s suggestion that the manuscript has to be more focused. As mentioned earlier, we reworked section 2 of the review, modified the introduction and we changed the title to SPMs “and” the lymphatic system instead of “in” to reflect the broader scope of the paper. Indeed, we geared our review article towards readers that may not be as familiar with either the lymphatics system or specialized pro-resolving mediators.

Round 2

Reviewer 1 Report

I am satisfied of the revised manuscript

Reviewer 3 Report

The authors have addressed my questions and concerns in the revised manuscript.